# Tumor and Nodal Disease Growth Rates in Patients with Oropharyngeal Squamous Cell Carcinoma

**DOI:** 10.3390/cancers15153865

**Published:** 2023-07-29

**Authors:** Nicole I. Farber, Yimin Li, Roberto N. Solis, Joy Chen, Zahrah Masheeb, Machelle Wilson, Arnaud F. Bewley, Marianne Abouyared, Shyam Rao, Yi Rong, Andrew C. Birkeland

**Affiliations:** 1Department of Otolaryngology-Head and Neck Surgery, University of California Davis, Sacramento, CA 95817, USAabewley@ucdavis.edu (A.F.B.);; 2Department of Radiation Oncology, University of California Davis, Sacramento, CA 94720, USAsdrao@ucdavis.edu (S.R.);; 3Department of Radiation Oncology, The First Affiliated Hospital of Xiamen University, School of Medicine, Xiamen University, Xiamen 361005, China; 4Davis-School of Medicine, University of California, Sacramento, CA 94720, USA; 5Department of Radiation Oncology, Mayo Clinic, Pheonix, AZ 85054, USA

**Keywords:** oropharyngeal, tumor growth, squamous cell carcinoma

## Abstract

**Simple Summary:**

Due to the overall favorable prognosis for p16+ oropharyngeal squamous cell carcinoma (OPSCC), de-escalation protocols are being implemented at a higher rate. However, there remains uncertainty regarding the best candidates for de-escalation therapy. Currently, tumor stage, p16 status, and history of smoking are widely accepted as influencing prognosis in OPSCC. Tumor and lymph node growth rates could potentially add to this growing list of prognosticators. Thus far, there is scant literature that defines tumor and nodal growth rate in OPSCC. In this study, we utilize radiation oncology software and interval CT scans to both define these rates and analyze their prognostic value. This analysis may offer a more comprehensive understanding of the natural course of OPSCC growth and aid head and neck surgeons, medical oncologists, and radiation oncologists in risk stratification.

**Abstract:**

Though specific growth rate (SGR) has potential prognostic value for oropharyngeal squamous cell carcinoma (OPSCC), there is sparse literature defining these rates. Our aims were to establish the SGRs of primary tumors (PTs) and lymph nodes (LNs) in OPSCC and to correlate SGR with oncologic outcome. A pilot study was designed with a retrospective analysis examining 54 patients from the University of California, Davis with OPSCC (diagnosed 2012–2019). Radiation oncology software and pretreatment serial CT scans were used to measure PT and LN volumes to calculate SGR and doubling time (DT). The mean PT-SGR was 1.2 ± 2.2%/day and the mean LN-SGR was 1.6 ± 1.9%/day. There was no statistically significant difference between slow-growing and fast-growing cohorts in terms of age, gender, smoking status, tumor subsite, HPV status (as determined with p16 staining), initial volume, or overall stage. SGR had no impact on 2-year overall survival, disease-free survival, or disease-specific survival. We found the average daily growth rates for OPSCC to be 1.2%/day and 1.6%/day. Our findings suggest PT- and LN-SGR are independent factors, not heavily influenced by known biomarkers and patient characteristics, without a statistical impact on prognosis. This information has value in patient counseling regarding tumor growth and in providing patients worried about fast-growing tumors the appropriate reassurance.

## 1. Introduction

Head and neck squamous cell carcinoma (HNSCC) remains a challenging disease to treat, particularly for aggressive subtypes. These cancers can have varying growth rates; however, in general they are considered fast-growing [1], which makes timely treatment critical. Delays in diagnosis to treatment lead to higher staging and potentially worse survival outcomes [2]. Specifically, in patients with oropharyngeal squamous cell carcinoma (OPSCC), tumor volume has been shown to be an important prognosticator for survival [3,4]. The median time to treatment initiation is around one month for patients with HNSCC; those with more advanced disease and/or being treated with chemoradiation often experience longer delays in care [5]. 

Despite this awareness of differing tumor growth rates, there are scant data that adequately define and characterize OPSCC pretreatment tumor growth rates. Furthermore, there are limited data investigating both primary cancer and metastatic nodal volume changes and survival outcomes. Tumor-specific growth rates in OPSCC have been associated with treatment failure and could potentially be used as a prognosticator, although studies remain limited [5]. Prognosis for OPSCC is generally favorable compared with other HNSCC subsites, particularly in p16+ OPSCC [6,7,8]. Due to this more favorable prognosis, de-escalation protocols are being used at a higher rate [9]. However, there remains uncertainty regarding the best candidates for de-escalation therapy, as some patients have proved to be poor responders to these protocols [10,11]. There is a critical need to identify patient and tumor characteristics and biomarkers that can predict poor responders to therapy. Currently, p16 status, tumor stage, and history of smoking are widely accepted as influencing prognosis in OPSCC [11]. Tumor and lymph node growth rates could potentially add to this growing list of prognosticators and assist in identifying patients who are appropriate for de-escalation protocols, or rather, intensification of treatment. 

Questions remain as to who is more likely to develop fast-growing tumors and whether they have worse clinical outcomes. It is not uncommon for patients to ask their providers for specifics regarding how much a tumor will grow daily or in a set timeframe. There is sparse literature that quantifies this rate. Understanding the natural course of OPSCC primary tumors and lymph nodes can aid providers in counseling patients and planning treatment [12]. Having a better knowledge of the growth patterns can assist in patient education and reassurance—two important factors that lead to a strong patient–provider relationship [13,14]. 

Herein, we aim to define the growth rates of OPSCC, assess the average daily growth rates of both primary-site and metastatic nodal disease, characterize the heterogeneity of tumor growth in this population, and to correlate primary cancer and metastatic nodal volume with oncologic outcomes. 

## 2. Materials and Methods

### 2.1. Patient Cohort

Fifty-four patients referred to the University of California, Davis Department of Otolaryngology—Head and Neck Surgery between 2012 and 2019 with biopsy-proven OPSCC (p16+ and p16−) were retrospectively evaluated. Inclusion criteria included histologically proven OPSCC and the availability of both a diagnostic CT scan and a serial CT scan before definitive treatment. Exclusion criteria included patients previously treated for head and neck cancer, prior radiotherapy (RT) applied to the head and neck region, and/or inadequate medical records. 

Definitive treatment was defined as initiation of RT with or without concurrent chemotherapy with curative intent based on tumor board recommendations and according to national guidelines [15]. Patients treated primarily with surgery but who had two scans with no interval therapy were included for tumor growth rate analysis, but not for oncologic outcome analysis. Data collected included demographics, p16 status, smoking history, and tumor stage. Tumors were staged according to the American Joint Committee on Cancer (AJCC) 8th edition. 

This study was conducted in accordance with the Declaration of Helsinki, and approved by the Institutional Review Board (or Ethics Committee) of University of California, Davis (protocol code: 1506220 and date of approval: 14 October 2019).

### 2.2. Imaging, Measurements, and Calculations

All CT scans were imported into MIM Maestro (MIM Software Inc, Cleveland, OH, USA). Two radiation oncologists (YL and SR) contoured the primary tumor (PT) and nodal disease in three dimensions to calculate volumes. Lymph nodes (LN) with minimal axial diameter >1 cm, a central necrosis >3 mm, or if present in neck levels close to the primary tumor in clusters of >3 were identified as pathological LNs [16]. All pathological LNs were contoured together and expressed as one volume called LN volume while the PT was expressed as PT volume.

Specific growth rate (SGR), defined as relative volume increase per unit of time, was calculated for PTs and LNs and labeled PT-SGR and LN-SGR, respectively. The equation SGR = ln(2nd volume/1st volume)/(t_2_ − t_1_) was used, which assumes growth to be exponential and defined as relative volume change given in percent per day, and is typically used to characterize tumor growth rate [17]. This measure uniformly estimates growth rates throughout all ranges. Additionally, doubling time (DT) for both PTs and LNs was calculated using the equation DT = (t_2_ − t_1_)ln2/ln(2nd volume/1st volume), which is expressed in days, and was used to compare our results with other studies [18]. Similar to previously cited methods by Murphy and colleagues, slow-growing and fast-growing cohorts were designated by utilizing the median SGR for the PT (1.2%/day) and LNs (1.3%/day) [19]. 

### 2.3. Statistical Analysis

Primary outcomes included disease-free survival (DFS), disease-specific survival (DSS), and overall survival (OS). All statistical analyses were performed using SAS^®^ software version 9.4 for Windows^®^ (SAS Institute Inc., Cary, NC, USA). Chi-square, Fisher exact, and one-way analysis of variance (ANOVA) tests were used to evaluate for significant differences. Kaplan–Meier curves and the log-rank test were performed to assess time-to-event differences, and logistic regression was used to evaluate differences in 2-year OS, DFS, and DSS. Disease-free survival was defined as complete and persistent elimination of disease. Disease-specific survival was defined as the percentage of patients who have not died from OPSCC. Follow-up was defined as the time between the initial scan and most recent follow-up appointment or date of death. We considered *p*-values < 0.05 to be statistically significant. 

## 3. Results

### 3.1. Patient Characteristics

Fifty-four patients met our inclusion criteria and were included in the study. There was one patient in which the primary tumor could not be clearly defined and therefore PT volume was not measured. Two patients had N0-stage necks; therefore, no LN volume was measured. Three patients treated primarily with surgery that had serial scans were included for PT volume and LN volume measurements, but oncologic outcome measures were not analyzed to maintain homogeneity in the treatment analysis. The average age among the 54 patients was 62.3 ± 9.4 years (mean ± standard deviation) and a majority of patients were men (83.3%) and former or current smokers (66.7%) (Table 1). The most common oropharyngeal tumor subsite was the base of the tongue (50%), followed closely by the tonsils (48.1%). Most patients had p16-positive tumors (92.5%) and stage I disease (40.7%), and underwent concurrent chemoradiation (CRT) as their definitive treatment (88.7%) (Table 1). 

### 3.2. Tumor Volumes, Growth Rates, and Doubling Time

The median interval time between the diagnostic CT and the planning CT was 38.5 (ranging from 5 to 158) days. The mean initial volume for the PT was 16.8 ± 12.6 mL with a mean final volume of 26 ± 18.9 mL. The mean volume of the LNs from the diagnostic CTs was 15.1 ± 15 mL with a mean volume of 23.9 ± 23.2 mL from the planning CTs. The mean increase in volume during the interval period was significant for both PTs and LNs (*p* = 0.0038 and 0.0237, respectively) (Table 2). The median SGR was similar for the PT cohort at 1.2%/day (25th percentile: 0.6; 75th percentile: 2.3) and the LN cohort at 1.3%/day (25th percentile: 0.6; 75th percentile: 2.2) (Table 2). The mean doubling times for the PT and LN cohorts were 71.7 ± 228.1 and 50.2 ± 286.4 days, respectively (Table 2).

### 3.3. Slow-Growing Versus Fast-Growing Tumors 

Patients were categorized into either the slow-growing PT cohort (PT-SGR < 1.19%/day) or the fast-growing tumor cohort (PT-SGR ≥ 1.19%/day) as well as the slow-growing LN cohort (LN-SGR < 1.28%/day) or the fast-growing LN cohort (LN-SGR ≥ 1.28%/day). There was no statistically significant difference between slow-growing and fast-growing PT cohorts in terms of age, gender, smoking status, tumor subsite, tumor p16 expression, overall stage, or the primary therapy they eventually received. This was also true for the slow-growing and fast-growing LN cohorts (Table 1). Additionally, there was no significant difference in the time between follow-ups for slow-growing and fast-growing cohorts. 

### 3.4. Factors Influencing SGR

The mean PT-SGR values for tumors of the tonsils, base of tongue, and soft palate were 2.2 ± 2.8%/day, 1.3 ± 1.2%/day, and 0.5 ± 0%/day, respectively. There was no statistically significant association between PT-SGR and the oropharyngeal subsites (*p* = 0.2551). The mean LN-SGR values for tumors of the tonsils, base of tongue, and soft palate were 2.0 ± 2.6%/day, 1.3 ± 1.1%/day, and 1.1 ± 0%/day, respectively (Table 3). Similar to PT-SGR, LN-SGR was not significantly associated with the primary tumor subsite (*p* = 0.4786). Both PT-SGR and LN-SGR were not significantly associated with the initial volume of the PT or LNs measured in the diagnostic CT (Table 3). Additionally, neither PT-SGR nor LN-SGR were significantly associated with p16 expression (*p* = 0.7021 and 0.7456, respectively) or overall stage (*p* = 0.300 and 0.753, respectively). Contrastingly, the time interval between the diagnostic CT and the planning CT significantly influenced SGR for both the PT and LNs (*p* = 0.0040 and 0.0042, respectively). The longer the time interval, the lower the observed PT-SGR and LN-SGR (Table 3). 

### 3.5. Survival Analysis

The median time to death and time to recurrence were stratified by slow- and fast-growing primary tumors and lymph nodes (Table 4). Growth rate did not have a significant effect on the time to death or recurrence in either of the primary tumor cohorts or lymph node cohorts. 

The 2-year OSs for slow-growing and fast-growing PTs were 84.6% and 92.6%, respectively (Figure 1a). This was comparable to the OSs for slow-growing and fast-growing LNs, which were 80.8% and 96.2%, respectively (Figure 2a). There was no significant difference between 2-year overall survival for fast-growing and slow-growing PTs or LNs (*p* = 0.37 and 0.12, respectively). Similar to OS, there was no significant difference in either DFS or DSS between slow-growing and fast-growing groups for the PT cohorts (Figure 1b,c) and LN cohorts (Figure 2b,c). 

## 4. Discussion

In this study, we identified the average SGRs of OPSCC primary tumors and lymph nodes to be 1.2%/day and 1.6%/day, respectively. We found that age, gender, smoking status, tumor subsite, HPV status, initial volume, and overall stage did not have a significant role in determining if a patient had a fast-growing versus slow-growing tumor. Additionally, we did not observe a correlation between SGR and survival. Although our study is limited in its power, this information has potential value in both patient counseling and treatment planning for OPSCC. 

Due to the overall favorable prognosis of p16+ OPSCC, de-escalation protocols are being implemented at a higher rate. However, a subset of these patients do not respond well to treatment and would not be appropriate for de-escalation therapy, but perhaps require intensified treatment regimens [11]. To date, there are no strong predictors of which patients may be poor responders. We investigated whether SGR may be a predictive biomarker in hopes of broadening the tools with which patients are stratified. Previous literature has demonstrated the prognostic value of OPSCC tumor volume and growth rates [3,19,20]. This is particularly important since traditional staging does not inform prognosis for HPV-positive OPSCC as it does with other SCCs of the head and neck [21]. The present analysis did not find PT-SGR or LN-SGR to be predictive biomarkers for survival in OPSCC. 

In order to compare OPSCC growth rates with other solid tumors, we calculated DT. The OPSCC mean PT-DT (71.7 days) was found to be greater than the HNSCC PT-DT (43 days) reported by Dejaco and colleagues, but lower than the HNSCC PT-DT (99 days) report by Jensen and colleagues [1,22]. Compared with other solid tumors such as breast cancer, lung carcinoma, and pancreatic adenocarcinoma, OPSCC demonstrates a more rapid growth (doubling time = 285 days, 181 days, and 144 days, respectively) [23,24,25]. To further delineate the growth patterns of OPSCC, SGR was calculated, as it has been shown to be a better marker of tumor growth rate and less affected by minor measurement uncertainties than DT [17]. The mean PT-SGR for OPSCC was 1.2%/day (median: 2.2%/day), which is less than the mean HNSCC PT-SGR of 1.8%/day reported by Dejaco et al. [1]. This was also true for Murphy and colleagues, who found a median OPSCC PT-SGR of 0.74%/day [19]. This finding suggests a less aggressive tumor progression of OPSCC when compared with other sites in the head and neck. In contrast to the PT, the median OPSCC LN-SGR (1.3%/day) was slightly greater than the HNSCC LN-SGR (1.2%/day) observed by Dejaco and colleagues [1]. 

Dejaco’s analysis allows us to compare our OPSCC growth rate with other sites in the head and neck. OPSCC primary tumors demonstrate a slower growth when compared with those in the oral cavity (2.4%/day) and hypopharynx (1.7%/day), but a faster growth when compared with PTs of the larynx (1.0%/day) [1]. Contrastingly, the median LN-SGR for OPSCC (1.3%/day) is slower than the rate in the hypopharynx (2.1%/day) but faster than the rates in the larynx (0.8%/day) and oral cavity (0.8%/day) [1]. Of note, the present analysis observed a PT-SGR that is slightly less than that of the LN-SGR. This may be explained by the characteristically small tumor size and high nodal dissemination observed in HPV-positive OPSCC, which a vast majority of our patient population expresses (92.5%) [26,27]. This characteristic of HPV may also explain the slowed PT-SGR demonstrated by OPSCC in comparison with other sites in the head and neck which are not considered HPV-mediated [28]. 

A major aim of our study was to determine who is more likely to develop fast-growing tumors and whether they have worse clinical outcomes. We utilized variables that are well-known prognostic factors in OPSCC, such as smoking history and p16 expression [27,28]. Despite their well-defined prognostic value, neither p16 status nor smoking status influenced whether a patient had a slow-growing or fast-growing tumor. However, this conclusion is limited by the small sample of patients with HPV-negative disease (*n* = 4), which is increasingly infrequent within the current clinical setting. Similarly, age, gender, tumor subsite, and overall stage did not have any influence on whether a patient belonged to the fast-growing cohort (Table 1). These findings further inform pretreatment risk stratification and provide clinicians with a more complete representation of the natural course of tumor behavior. 

In a similar analysis, we found that OPSCC PT-SGR and LN-SGR were independent of tumor subsite, p16 status, initial PT volume, initial LN volume, and overall stage (Table 3). This differs from other HNSCC lymph node behaviors as observed by Dejaco and colleagues, who found the HNSCC LN-SGR to be influenced by initial LN volume and tumor subsite, specifically the larynx [1]. Similar to our analysis, Dejaco et al. found no observed influence of initial volume on SGR within the oropharyngeal subsite [1]. Contrastingly, Murphy and colleagues demonstrated a significantly increased PT-SGR in p16-negative OPSCC tumors and tumors with larger volumes [19]. Again, these conflicting results may be secondary to the relatively small patient population with HPV-negative tumors in our study (*n* = 4; 7.5%). 

The previous literature has established a clear negative association between treatment waiting time and outcomes in HNSCC [19,22,29,30]. Murphy and coauthors calculated 46 to 52 days to be the threshold, after which patients have decreased OS [30]. The median time between the diagnostic CT and planning CT for the studied population was 38.5 days (ranging from 5 to 158). This is similar to previously reported intervals and lies under the threshold [1,19]. The present analysis found that the longer the interval between diagnostic CT scans and planning CT scans, the significantly lower the OPSCC PT-SGR and LN-SGR (*p* = 0.0040 and 0.0042, respectively). This was also demonstrated by Dejaco and colleagues for LN-SGR in HNSCC, but not PT-SGR [1]. This suggests that there is not a significant interval growth between diagnosis and treatment scans. This result supplements our current understanding of OPSCC tumor behavior and aids otolaryngologists, radiation oncologists, and medical oncologists in understanding the long-term impact of treatment timing. This aids in both patient counseling for appropriate treatment timing and reassuring patients on realistic timeframes. 

A key aim of our study was determining if pretreatment growth rate is an important prognosticator for OPSCC. Our survival analysis demonstrated that neither PT-SGR nor LN-SGR had an impact on OS, DFS, or DSS in OPSCC. This aligns with prior analyses, specifically that of Dejaco et al., which demonstrated that SGR has no impact on OS in all HNSCCs [1]. Alternatively, Davis and colleagues identified initial lymph node volume to be negatively associated with DFS in p16-positive OPSCC [3]. This suggests that, perhaps, initial volume is a better prognosticator than pretreatment growth rate. It is important to note, again, that conclusions drawn from our analysis regarding OPSCC behavior are limited by the disproportionate quantity of HPV-positive tumors, which are a biochemically and clinically distinct entity. As the benefits of deintensified therapy come to light, risk stratification remains a critical component of oncologic care for patients with OPSCC. This study builds a scaffold on which larger, multi-institutional analyses can be built to help answer this ongoing clinical controversy. 

Several limitations in the present study should be acknowledged. The current analysis carries the inherent limitations of a retrospective review and small sample size. Additional studies in a larger, multi-institutional review fashion would be additive. Most patients included in this analysis underwent radiation therapy as definitive treatment, and thus tend to have more complex or advanced tumors than surgical candidates. Nonetheless, our survival rates may be skewed based on this characteristic of the patients. To better strengthen the associations identified within our analysis, future studies should aim to stratify patients based on p16 status. And lastly, our analysis is limited by inherent human error that could have occurred during tumor contouring. Continued advancements in imaging and artificial intelligence can aid in decreasing this component of our analysis. 

## 5. Conclusions

As the emphasis on de-escalation therapy in OPSCC increases, the need to identify biomarkers to stratify patients has become readily apparent. We identified the average OPSCC SGRs of the primary tumor and lymph nodes to be 1.2%/day and 1.6%/day, respectively. We did not find that age, gender, smoking status, tumor subsite, p16 status, initial volume, or overall stage had a significant role in determining if a patient had a fast-growing versus slow-growing tumor. Importantly, we did not observe a correlation between SGR and 2-year overall survival, disease-free survival, or disease-specific survival. The current analysis adds to the growing literature that suggests growth rate to be an independent characteristic, not heavily influenced by known biomarkers and patient characteristics, and an overall poor prognosticator. This information allows us to better understand the natural course of OPSCC, which in turn will assist providers in both stratifying their patients for the most appropriate treatment and counseling them on expected outcomes. 

## Figures and Tables

**Figure 1 cancers-15-03865-f001:**
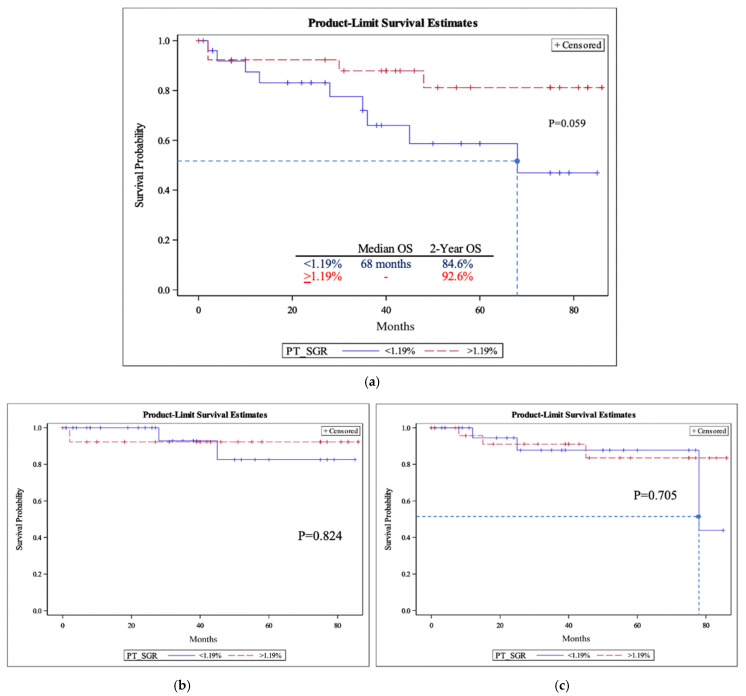
The primary tumor-specific growth rate (PT-SGR) overall survival (**a**), disease-specific survival (**b**), and disease-free survival (**c**). (**a**) Primary tumor overall survival (OS). Median overall survival and two-year overall survival for the two groups are included. The dashed line corresponds to the median time. *p*-values were obtained using the log-rank test. Logistic regression for two-year overall survival had a *p*-value of 0.37. (**b**) Primary tumor specific growth rate (PT-SGR) disease specific survival (DSS). *p*-values were obtained using the log-rank test. (**c**) Primary tumor specific growth rate (PT-SGR) disease free survival (DFS). The dashed line corresponds to the median time. *p*-values were obtained using the log-rank test.

**Figure 2 cancers-15-03865-f002:**
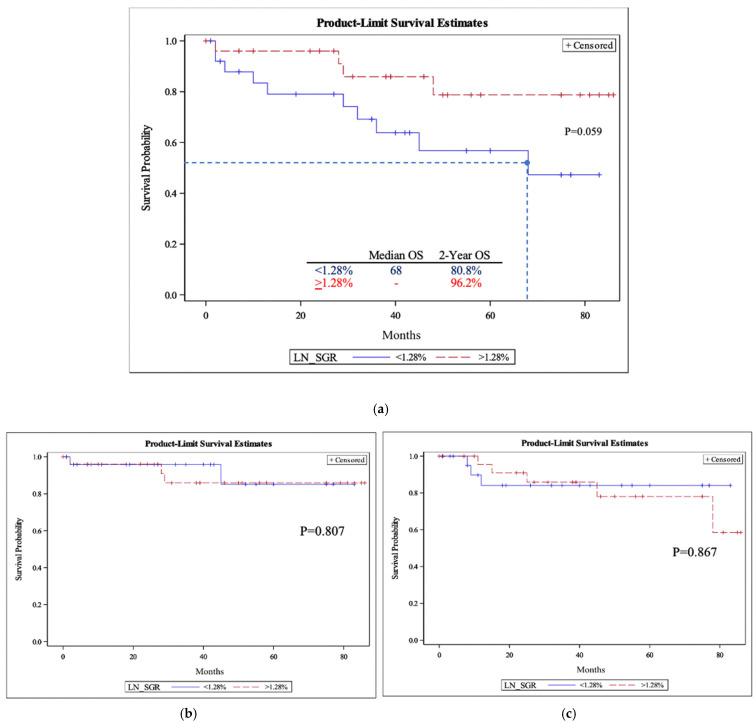
The lymph node-specific growth rate (LN-SGR) overall survival (**a**), disease-specific survival (**b**), and disease-free survival (**c**). (**a**) Lymph node-specific growth rate (LN-SGR) overall survival (OS). Median overall survival and two-year overall survival for the two groups are included. The dashed line corresponds to the median time. *p*-values were obtained using the log-rank test. Logistic regression for two-year overall survival had a *p*-value of 0.12. (**b**) Lymph node specific growth rate (LN-SGR) disease specific survival (DSS). *p*-values were obtained using log-rank test. (**c**) Lymph node specific growth rate (LN-SGR) disease specific survival (DFS). *p*-values were obtained using log-rank test.

**Table 1 cancers-15-03865-t001:** A comparison of baseline patient characteristics between fast- and slow-growing cohorts.

	All Patients (*n* = 54)	PT-SGR < 1.19% (*n* = 26)	PT-SGR ≥ 1.19% (*n* = 27)	*p*-Value	LN-SGR < 1.28% (*n* = 26)	LN-SGR ≥ 1.28% (*n* = 28)	*p*-Value
**Age (Years)**							
Mean ± SD	62.3 ± 9.4	60.2 ± 9.8	64.7 ± 8.1	0.0813	62 ± 9.4	63 ± 9.5	0.6993
**Gender (*n*, %)**							
Male	45 (83.3)	19 (73)	25 (92.6)		21 (80.7)	24 (85.7)	
Female	9 (16.6)	7 (26.9)	2 (7.4)	0.0764	5 (19.2)	4 (14.3)	0.7237
**Smoking (*n*, %)**							
Never	18 (33.3)	7 (26.9)	11 (40.7)		9 (34.6)	9 (32.1)	
Former	26 (48.1)	14 (53.8)	12 (41.7)		14 (53.8)	12 (42.8)	
Current	10 (18.5)	5 (19.2)	4 (14.8)	0.8395	3 (11.5)	7 (25)	0.4317
**Subsite (*n*, %)**							
Tonsil	26 (48.1)	10 (38.5)	15 (55.6)		10 (38.5)	16 (57.1)	
Base of tongue	27 (50)	15 (57.7)	12 (44.4)		15 (57.7)	12 (42.9)	
Soft palate	1 (1.9)	1 (3.8)	0 (0)	0.3135	1 (3.8)	0 (0)	0.2658
**Stage (*n*, %)**							
I	22 (41)/20 (38)	9 (35)	13 (48)	0.328	12 (46)	8 (31)	0.446
II	17 (32)/17 (33)	9 (35)	8 (30)		6 (23)	11 (42)	
III	11 (21)/12 (23)	5 (19)	6 (22)		7 (27)	5 (19)	
IV	3 (6)/3 (6)	3 (11)	0 (0)		1 (4)	2 (8)	
**P16 Status (*n*, %) ***							
Positive	49 (92.5)	23 (88.5)	26 (96.3)		24 (92.3)	25 (92.6)	
Negative	4 (7.5)	3 (11.5)	1 (3.7)	0.3507	2 (7.7)	2 (7.4)	>0.99
**Primary Therapy** **(*n*, %) ***							
RT	5 (9.4)	1 (3.8)	4 (14.8)		1 (3.8)	4 (14.8)	
CRT	47 (88.7)	24 (92.3)	23 (85.2)	0.3517	25 (96.2)	22 (81.5)	0.3497
**Follow-Up (Months)**							
Mean ± SD	39 ± 28.6	33.9 ± 26.6	44.9 ± 28.9		35.8 ± 27.0	40.5 ± 28.9	
Median (range)	37.5 (1–86)	33.5 (1–85)	39.5 (2–86)	0.157	22.5 (1–77)	37.5 (2–86)	0.548

* There is one patient missing information in this category, making the total *n* = 53.

**Table 2 cancers-15-03865-t002:** Primary tumor and lymph node volume and growth rates.

	Initial Volume (mL)	Final Volume (mL)	Specific Growth Rate (%/day)	Doubling Time (days)
Primary tumor (mean ± SD)	16.8 ± 12.6	26 ± 18.9	1.2 ± 2.2%	71.7 ± 228.1
(Median, 25th; 75th percentile)			1.2 (0.6; 2.3)	36.5 (17.4; 74.5)
Lymph node (mean ± SD)	15.1 ± 15	23.9 ± 23.2	1.6 ± 1.9%	50.2 ± 286.4
(Median, 25th; 75th percentile)			1.3 (0.6; 2.2)	38.9 (21.6; 91.5)

**Table 3 cancers-15-03865-t003:** The roles of different variables on the specific growth rate of the primary tumor and lymph nodes.

	N	PT-SGR (%/Day)	*p*-Value	LN-SGR (%/Day)	*p*-Value
**Tumor Subsite**					
**Tonsils**	26	2.2 ± 2.8		2.0 ± 2.6	
**Base of tongue**	27	1.3 ± 1.2		1.3 ± 1.1	
**Soft palate**	1	0.5 ± 0	0.2551	1.1 ± 0	0.4786
**P16 Status**					
**Positive**	50	1.7 ± 2.2		1.65 ± 2.0	
**Negative**	4	1.3 ± 1.1	0.7021	1.32 ± 0.6	0.7456
**Time Between Scans (Days)**					
**0–25**	16	3.4 ± 3.2		2.93 ± 3.0	
**26–50**	24	1.1 ± 1.0		1.14 ± 0.8	
**>50**	14	0.8 ± 0.5	0.0040	1.00 ± 0.9	0.0042
**Initial PT Volume**					
**<15 mL**	30	2.0 ± 2.8		1.70 ± 2.4	
**≥15 mL**	24	1.3 ± 0.9	0.2322	1.53 ± 1.2	0.7530
**Initial LN Volume**					
**<15 mL**	32	2.1 ± 2.6		1.9 ± 2.3	
**≥15 mL**	22	1.1 ± 1.1	0.0954	1.2 ± 1.1	0.1912
**Tumor Stage**					
**I**	20/22	1.3 ± 0.9		1.8 ± 1.8	
**II**	17	2.3 ± 3.0	0.300	1.9 ± 3.1	0.753
**III**	11/12	1.1 ± 1.0		1.3 ± 0.9	
**IV**	3	1.6 ± 0.5		0.8 ± 2.1	

**Table 4 cancers-15-03865-t004:** Time to death and recurrence stratified by slow- and fast-growing lymph nodes and primary tumors.

		Time to Death		Time to Recurrence		
	*n*	Mean (SD)	*p*-Value	*n*	Mean (SD)	*p*-Value
LN-SGR						
<1.28%	10	24.1 (21.8)	0.836	3	9.7 (2)	0.177
>1.28%	4	26.8 (18.9)		5	34.8 (27.5)	
PT-SGR						
<1.19%	9	26.8 (21.8)	0.644	3	38.3 (34.9)	0.536
>1.19%	4	20.5 (22.6)		3	22.7 (19.7)	

*p*-values were obtained using independent Student’s *t*-test. Abbreviations: LN-SGR: lymph node-specific growth rate; PT-SGR: primary tumor-specific growth rate.

## Data Availability

Data are stored at UC Davis and available upon request.

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
