# Peer review of "Tumor and Nodal Disease Growth Rates in Patients with Oropharyngeal Squamous Cell Carcinoma"

_cancers, 2023, doi:10.3390/cancers15153865_

Round 1
Reviewer 1 Report
Dear Authors,
I read with interest your work "Tumor and Nodal Disease Growth Rates in Patients with Oro-pharyngeal Squamous Cell Carcinoma" that discuss an interesting topic.
The authors aimed to define the growth rates of OPSCC, to assess the average daily growth rates of both primary site and metastatic nodal disease, characterize the heterogeneity of tumor growth in this population, and to correlate primary cancer and metastatic nodal volume with oncologic outcomes.
While the results and conclusions are not novel, they are based on solid statistical data and help to extend the knowledge on the causes of adipose cell resorption, which may be beneficial in the future.
Therefore, I have no further comments against the manuscript.
Reviewer 2 Report
It was not mentioned in the text (Material and Methods) the Ethics approval number and date.
A higher clarity and concision may be useful. In the Conclusion it is not necessary to mention the aim of the study. It was already presented. The specific conclusions based on the results, may be necessary.
'This study aimed to identify a potential biomarker, quantify growth rates for OPSCC, and to evaluate if it has any association with other known biomarks or survival''.
A reading of the text and correction of writing and editing small errors may be useful.
